# Post-Laryngectomy Voice Prosthesis Changes by Speech-Language Pathologists: Preliminary Results

**DOI:** 10.3390/jcm11144113

**Published:** 2022-07-15

**Authors:** Stéphane Hans, Grégoire Vialatte de Pemille, Robin Baudouin, Aude Julien-Laferriere, Florent Couineau, Lise Crevier-Buchman, Marta P. Circiu, Jérôme R. Lechien

**Affiliations:** 1Department of Otorhinolaryngology and Head and Neck Surgery, Foch Hospital, School of Medicine, UFR Simone Veil, Université Versailles Saint-Quentin-en-Yvelines (Paris Saclay University), F-92150 Paris, France; prhans.foch@gmail.com (S.H.); g.vialatte-de-pemille@hopital-foch.com (G.V.d.P.); robin.baudouin@aol.fr (R.B.); ajlaferriere@gmail.com (A.J.-L.); florentcouineau@gmail.com (F.C.); lise.buchman1@gmail.com (L.C.-B.); mp.circiu@gmail.com (M.P.C.); 2Department of Human Anatomy and Experimental Oncology, Faculty of Medicine, UMONS Research Institute for Health Sciences and Technology, University of Mons (UMons), B-7000 Mons, Belgium; 3Department of Otorhinolaryngology and Head and Neck Surgery, Elsan Polyclinic of Poitiers, F-86000 Poitiers, France

**Keywords:** total laryngectomy, cancer, voice, voice prosthesis, otolaryngology, head neck surgery, speech language therapists

## Abstract

Background: In the present study, we assess the feasibility and success outcomes of voice prosthesis (VP) changes when performed by a speech-language pathologist (SLP). Methods: Patients treated with total laryngectomy (TL) from January 2020 to December 2020 were prospectively recruited from our medical center. Patients benefited from tracheoesophageal puncture. The VP changes were performed by the senior SLP and the following data were collected for each VP change: date of placement; change or removal; VP type and size; reason for change or removal; and use of a washer for periprosthetic leakage. A patient-reported outcome questionnaire including six items was proposed to patients at each VP change. Items were assessed with a 10-point Likert-scale. Results: Fifty-two VP changes were performed by the senior SLP during the study period. The mean duration of the SLP consultation, including patient history, examination and VP change procedure, was 20 min (range: 15–30). The median prosthesis lifetime was 88 days. The main reasons for VP changes were transprosthetic (*n* = 34; 79%) and periprosthetic (*n* = 7; 21%) leakages. SLP successfully performed all VP changes. He did not change one VP, but used a periprosthetic silastic to stop the periprosthetic leakages. In two cases, SLP needed the surgeon’s examination to discuss the following indication: implant mucosa inclusion and autologous fat injection. The patient satisfaction was high according to the speed and the quality of care by the SLP. Conclusions: The delegation of VP change from the otolaryngologist–head and neck surgeon to the speech-language pathologist (SLP) may be achieved without significant complications. The delegation of VP change procedure to SLP may be interesting in some rural regions with otolaryngologist shortages.

## 1. Introduction

Total laryngectomy (TL) is a common oncological surgery in head and neck surgery. The post-TL voice rehabilitation is challenging for both patients and practitioners due to the complex nature of patient presentation and the involvement of many motivational and oncological factors [1,2]. To date, tracheoesophageal speech is considered the gold standard for post-TL voice rehabilitation [1,2]. The mean voice prosthesis (VP) lifetime ranges from 3 to 6 months, which supports the need of adequate follow-up and VP changes 3. In most countries, the VP changes are performed by physicians because it is considered as a medical act.

In the present study, we assessed the feasibility and success outcomes of VP changes when performed by a speech-language pathologist (SLP).

## 2. Materials and Methods

### 2.1. Ethical Consideration

The local institutional review board approved the study protocol (APHP-HEGP-2018). A waiver of informed consent of participants was granted because participant data were protected and anonymized.

### 2.2. Subjects and Setting

Patients treated with TL from January 2020 to December 2020 were prospectively recruited from our medical center. Patients benefited from a tracheoesophageal puncture and 1-month post-TL VP. The surgeon used the Provox^®^ 2 type prosthesis (Atos Medical AB, Hörby, Sweden). Patients were followed by an experienced otolaryngologist and SLP for the voice rehabilitation and the oncological follow-up. The first VP change was performed by the senior SLP (GD) who was supervised by the senior head and neck surgeon (SH). The rest of the VP changes were performed by the same SLP without surgeon supervision. However, the surgeon was called in the case of problems.

### 2.3. Practitioner and Patient Outcomes

The following outcomes were considered: gender; age; primary tumor site; cTNM classification; primary treatment; TL indication (primary, salvage, second primary, and dysfunctional larynx); surgical characteristics (e.g., neck dissection and flap reconstruction); driving distance to the hospital; and survival outcome. The following data were collected for each VP change: date of placement; change or removal; VP type and size; reason for change or removal; and use of a washer for periprosthetic leakage.

A patient-reported outcome questionnaire including 6 items was proposed to patients at each VP change. Items were assessed with a 10-point Likert-scale.

## 3. Results

Ten patients completed the evaluations. The epidemiological and clinical outcomes of patients are available in Table 1. There were eight males and two females, respectively. The median age was 63.2 yo (range of 48–79 yo). TL was performed for the following indications: low-grade cricoid chondrosarcoma (*n* = 2), recurrent laryngeal cancer after radiation (*n* = 3), or chemoradiation (*n* = 5).

Fifty-two VP changes were performed by the senior SLP during the study period. The mean duration of the SLP consultation, including patient history, examination, and VP change procedure, was 20 min (range: 15–30). The median prosthesis lifetime was 88 days. The main reasons for VP changes were transprosthetic (*n* = 34; 79%) and periprosthetic (*n* = 7; 21%) leakages. SLP successfully performed all VP changes. He did not change one VP, but used a periprosthetic silastic to stop the periprosthetic leakages. In two cases, the SLP needed the surgeon’s examination to discuss the following indications: implant mucosa inclusion and autologous fat injection.

The patient satisfaction was high according to the speed and the quality of care by the SLP (Table 2).

## 4. Discussion

Voice rehabilitation after TL is an important postoperative issue for the patient quality of life [3,4,5]. In practice, the VP change is a simple procedure that is usually performed by residents or board-certified physicians. In this study, we reported adequate SLP and patient-reported outcome perception about the SLP-related VP change. The delegation of some clinical tasks from the otolaryngologist–head and neck surgeon to the SLP is a current topical issue that may be associated with many advantages.

First, it is commonly accepted that the development of post-TL tracheoesophageal speech involves important speech rehabilitation work and adequate follow-up for the management of VP leakage, which may be time-consuming for the physician [4]. Currently, the number and the availability of otolaryngologists in rural areas may be limited in some European regions regarding some government hospital reforms that led to significant reductions in medical centers and physicians [6,7]. In our country, the shortage of otolaryngologists in rural regions may lead to patient proposition of post-TL esophageal speech rather than tracheoesophageal speech to limit the need of post-TL care [8]. In that way, the availability of SLPs in the management of VP changes may, therefore, be an advantage for the patient accessibility to health care and follow-up. Second, in some world regions, SLPs already perform routine videolaryngostroboscopy, which was associated with the enhancement of the SLP role in the decision-making process in voice restoration [9]. According to the voice rehabilitation process, SLPs know their patients well, and a trusting relationship may develop throughout the rehabilitation sessions. In the present study, more than 90% of patients reported a high rate of satisfaction outcomes about the SLP-VP change procedure, which may be explained by the trusting relationship between the SLP and patient and the feasibility of the procedure.

The delegation of VP changes to SLP makes particular sense in our country because SLPs have been able to prescribe respiratory or phonatory rehabilitation equipment for TL patients for the past 4 years (law of 30 March 2017). Interestingly, a recent Italian study reported that physicians were not opposed to the delegation of this task to other health professionals, which strengthens the need of debate about this task delegation issue.

The primary limitations of the present study are the low number of procedures performed by the SLP (42 procedures) and the low number of patients, which limited the realization of statistical analysis. The lack of use of validated patient-reported outcome questionnaires assessing the VP change procedure is an additional limitation. To the best of our knowledge, there is no similar study available in the literature, which is the main strength of this preliminary study.

## 5. Conclusions

The VP change is a feasible procedure for SLP associated with few complications, rare need of physician intervention and adequate patient-reported outcome perception. Future controlled studies are needed to compare VP change outcomes between physicians and SLPs and to evaluate its cost-effectiveness.

## Figures and Tables

**Table 1 jcm-11-04113-t001:** Characteristics of patients followed by the speech therapist.

Patient Number	Age (year)	Gender	Comorbidities	Initial Treatment	Indications	cTNM	VP (nb)	Complications
1	75	F	Tobacco	RT	Rec. LSCC	T3N0	4	-
2	79	M	Tobacco	RT	Rec. LSCC	T2N0	3	-
3	64	M	Tobacco, HTA	CRT	Rec. LSCC	T3N1	5	-
4	58	M	HTA	-	CS	-	7	-
5	61	F	Tobacco	CRT	Rec. LSCC	T3N0	4	-
6	68	M	Tobacco	CRT	Rec. LSCC	T2N1	4	-
			HTA					
7	57	M	Tobacco	RT	Rec. LSCC	T1N0	4	-
8	48	M	Tobacco	CRT	Rec. LSCC	T3N0	5	-
9	52	M	Tobacco	CRT	Rec. LSCC	T3N0	3	-
10	70	M	-	-	CS	-	4	-

Abbreviations: CS = chondrosarcoma; RT: radiation therapy; CRT = chemoradiation; F/M = female/male; HTA = hypertension; m = minutes; Rec. LSCC = recurrent laryngeal squamous cell carcinoma; VP = Voice prosthesis; nb: number of prosthesis during the study period.

**Table 2 jcm-11-04113-t002:** Responses to questionnaires.

Questions/Answers	1	2–3	4–5	6–7	8–9	10
Early appointement	39 (93)	2 (5)	1 (2)	-	-	-
Speed and availability of practitioner	39 (93)	2 (5)	1 (2)	-	-	-
Quality of care	40 (96)	1 (2)	1 (2)			
Voice prosthesis change speed	39 (93)	-	2 (5)	-	1 (2)	-
Discomfort during change	32 (75)	4 (10)	4 (10)		2 (5)	
Speech therapist for voice prosthesis change in the future	40 (95)	1 (5)	1 (5)	-	-	-

The numbers in brackets are %. Forty-two patients completed a 10-point evaluation of quality and speed of care, ranging from 1 (very high satisfaction) to 10 (very low satisfaction).

## Data Availability

Data are available upon request.

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
