# Peer review of "Post-Laryngectomy Voice Prosthesis Changes by Speech-Language Pathologists: Preliminary Results"

_jcm, 2022, doi:10.3390/jcm11144113_

Round 1
Reviewer 1 Report
Hans and colleagues evaluated the feasibility and success outcomes of voice prosthesis (VP) changes when performed by a speech-language pathologist (SLP). I find the work interesting and very applicable, and my only complaint is the small number of patients and the relatively small number of voice prosthesis changes.
Congratulations to the authors.
Author Response
Thank you. Yes, we agree. This is a feasibility study and we submitted it as Brief Report considering these limitations.

Reviewer 2 Report
The authors present an interesting study on the role of Speech-Language-Pathologists in the substitution of voice protesis in laryngectomized patients. They demonstrated that the SLP can manage this task with general patient satisfaction. It must be considered that these are preliminary results, but in this case I think that more numbers can only improve the patient's satisfaction, because as every procedure, more is better. Good discussion, particularly I agree on the need to organize and distribute the work among the different specialists to allow better care access to patients
I’ve only some comments and suggestions:
1- line 1: “Total laryngectomy (TL) is one of the most common procedures in head and neck surgery”. I think this sentence is a bit strong. If we're speaking about "demolition" oncological head and neck surgery, I can agree, but if we speak about "general" ENT surgery, I can't. I invite the author to rewrite this opening sentence or to present a dedicated reference to better support their position.
Table 2: specify the number in brackets are the percentages. In line 3, due to rounding, the result is 99
Generally speaking, I suggest a complete English review, to make the text easier to read.
For example…
2 - line 5-6: “To date, tracheoesophageal speech is considered as the gold standard for post-TL voice rehabilitation”. I suggest to remove “as”
3-line 7 “The mean duration of voice prosthesis (VP) life-time ranged from 3 to 6-month” . I think it is better as follows: “The mean voice prosthesis (VP) life-time ranges from 3 to 6-months, “
4 -paragraph 2.2 : “the surgeon being however called in case of problems” → “ However, the surgeon was called in case of problems “
Author Response
Reviewer 2:
The authors present an interesting study on the role of Speech-Language-Pathologists in the substitution of voice protesis in laryngectomized patients. They demonstrated that the SLP can manage this task with general patient satisfaction. It must be considered that these are preliminary results, but in this case I think that more numbers can only improve the patient's satisfaction, because as every procedure, more is better. Good discussion, particularly I agree on the need to organize and distribute the work among the different specialists to allow better care access to patients
Thank you. Yes, we agree. This is a feasibility study and we submitted it as Brief Report considering these limitations.
I’ve only some comments and suggestions:
1- line 1: “Total laryngectomy (TL) is one of the most common procedures in head and neck surgery”. I think this sentence is a bit strong. If we're speaking about "demolition" oncological head and neck surgery, I can agree, but if we speak about "general" ENT surgery, I can't. I invite the author to rewrite this opening sentence or to present a dedicated reference to better support their position.
We rewrote the sentence as: Introduction, p.1, line 1: “Total laryngectomy (TL) is a common oncological surgery in head and neck surgery.”
Table 2: specify the number in brackets are the percentages. In line 3, due to rounding, the result is 99
We added in the footnotes: “Total laryngectomy (TL) is a common oncological surgery in head and neck surgery.”
We corrected: 40 (96), 1 (2), 1 (2): total of 100%
Generally speaking, I suggest a complete English review, to make the text easier to read.
For example…
2 - line 5-6: “To date, tracheoesophageal speech is considered as the gold standard for post-TL voice rehabilitation”. I suggest to remove “as”
We modified. P.1, line 5: “To date, tracheoesophageal speech is considered the gold standard for post-TL voice rehabilitation [1,2].”
3-line 7 “The mean duration of voice prosthesis (VP) life-time ranged from 3 to 6-month” . I think it is better as follows: “The mean voice prosthesis (VP) life-time ranges from 3 to 6-months, “
We modified: P.1, line 7: “The mean voice prosthesis (VP) life-time ranges from 3 to 6-month, which supports the need of adequate follow-up and VP changes 3.”
4 -paragraph 2.2 : “the surgeon being however called in case of problems” → “ However, the surgeon was called in case of problems “
We modified: P.2, 2.2., last line: “The rest of the VP changes were performed by the same SLP without surgeon supervision. However, the surgeon was called in case of problems.”
Moreover, we check the spelling/grammar a second time with a native EN speaker.
Thanking you in advance for your attention, I remain,
Best regards,
